# RGB-Event ISP: The Dataset and Benchmark

**Yunfan Lu, Yanlin Qian, Ziyang Rao, Junren Xiao, Liming Chen[2], Hui Xiong**[*]
AI Thrust, Hong Kong University of Science and Technology (Guangzhou);  AlpsenTek[2]
ylu066@connect.hkust-gz.edu.cn, qianyanlin619812051@gmail.com
{zrao538,jxiao767}@connect.hkust-gz.edu.cn
liming.shen@alpsentek.com, xionghui@ust.hk

## Abstract

Event-guided imaging has received significant attention due to its potential to revolutionize instant imaging systems. However, the prior methods primarily focus on enhancing RGB images in a post-processing manner, neglecting the challenges of image signal processor (ISP) dealing with event sensor and the benefits events provide for reforming the ISP process. To achieve this, we conduct the first research on event-guided ISP. First, we present a new event-RAW paired dataset, collected with a novel but still confidential sensor that records **pixel-level aligned** events and RAW images. This dataset includes 3373 RAW images with $2248 \times 3264$ resolution and their corresponding events, spanning 24 scenes with 3 exposure modes and 3 lenses. Second, we propose a conventional ISP pipeline to generate good RGB frames as reference. This conventional ISP pipleline performs basic ISP operations, *e.g.* demosaicing, white balancing, denoising and color space transforming, with a ColorChecker as reference. Third, we classify the existing learnable ISP methods into 3 classes, and select multiple methods to train and evaluate on our new dataset. Lastly, since there is no prior work for reference, we propose a simple event-guided ISP method and test it on our dataset. We further put forward key technical challenges and future directions in RGB-Event ISP. In summary, to the best of our knowledge, this is the very first research focusing on event-guided ISP, and we hope it will inspire the community. The code and dataset are available at: https://github.com/yunfanLu/RGB-Event-ISP.

## 1 Introduction

Since their invention in 1975, digital cameras have profoundly impacted various aspects of modern society (Delbracio et al., 2021; Kyung et al., 2016). Active pixel sensors (APS) (Liebe et al., 1998) are used as the core of cameras to capture RGB color signals, recording images or videos. This technology forms the foundation for widespread applications in smartphones (Delbracio et al., 2021), autopilot systems (Ingle & Phute, 2016), drones (Zhu et al., 2018), virtual reality (Huang et al., 2017), and more. However, nowadays APS has reached a bottleneck *wrt.* power consumption, frame rate, and dynamic range due to its global recording characteristics (Gallego et al., 2020). Event vision sensors (EVS), with their inherent asynchronous recording property, achieve lower power consumption ($< 10mW$), lower latency ($< 1ms$), and higher dynamic range ($> 120dB$) (Gallego et al., 2020). As a result, integrating EVS as a significant enhancement to APS imaging system has received considerable attention in recent years (Lu et al., 2023b; Tulyakov et al., 2021; Gallego et al., 2020; Tulyakov et al., 2022). Heavy efforts have been put on developing new imaging system combining EVS and APS (Shariff et al., 2024; Lu et al., 2023b;a). The introduction of EVS has nearly reshaped the entire framework of imaging formation and enhancement, impacting almost all relevant areas *e.g.*, video super-resolution (Lu et al., 2023b; Jing et al., 2021), video frame interpolation (Tulyakov et al., 2021; 2022; Lu et al., 2023a), deblurring (Yuan et al., 2007; Zhang et al., 2022; Yunfan et al., 2023), high dynamic range imaging (Xiaopeng et al., 2024; Messikommer et al., 2022), low-light image enhancement (Wang et al., 2020b; Liang et al., 2024), and rolling shutter correction (Zhou et al., 2022; Lu et al., 2023a). *However, the majority of previous work focuses on using events as auxiliary information to boost the performance of classical RGB imaging*

---

[*]corresponding author

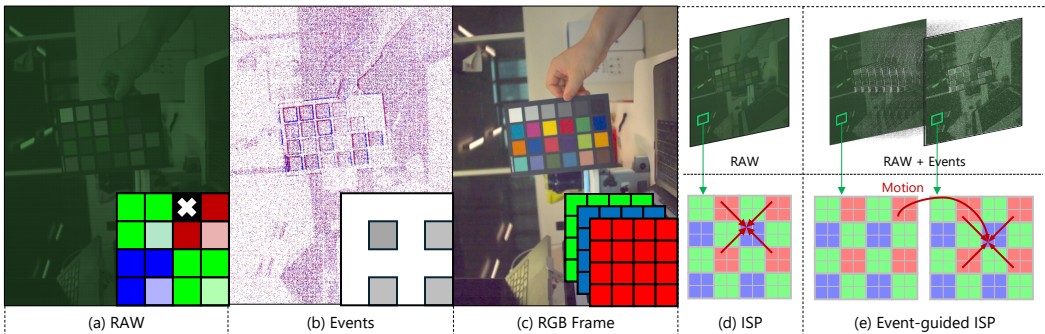

Figure 1: (a), (b), and (c) display a RAW, Events, and RGB frame captured by the hybrid vision sensor (HVS), respectively. The RAW image follows a quad-Bayer pattern (Yang et al., 2022), while the events are positioned at the lower-right corner of each color pixel block, making the RAW resolution twice that of the events. (d) illustrates the traditional ISP process. (e) shows the potential event-guided ISP process, where the higher temporal resolution of events can captures motion information for ISP.

*systems, while methods and benchmarks that considering the challenges and opportunities of events in the APS ISP process, are lacking.*

Merging APS and EVS in ISP is non-trivial on the implementation level. Prism spectrometer is an early stage attempt and it needs the corresponding optical mechanic setting (Tulyakov et al., 2022). However, this prism-based approach is very cumbersome, requiring additional optical prisms and failing to ensure the alignment between APS and EVS. Sensors that integrate both APS and EVS on the photodiode level are referred to as hybrid-vision sensors (HVS) (Yaqi et al., 2024; MIPI Challenge 2024, 2024), which represent a cutting-edge technology, offering significant advancements in camera imaging. Due to the manufacturing complexity and error-prone design process of HVS, the RAW data generated by APS in HVS exhibits higher noise, missing values at fixed positions, and is more sensitive to defects (MIPI Challenge 2024, 2024; Yaqi et al., 2024). Recent works have acknowledged this challenge and proposed datasets for demosaicing, denoising, or defect correction for APS RAW, where *the challenges in APS of HVS take precedence over the potential benefits events signal could provide.* With the inherent higher dynamic range and lower latency, events can perceive a broader spectrum and capture more-instant motion information (Shekhar Tripathi et al., 2022; Liang et al., 2021), allowing significant potential for boosting the denoising and color correction of ISP processing of APS RAW, as shown in Fig. 1.

To better explore the benefits of events on the ISP process of HVS, we propose a new dataset with **pixel-wise aligned** events and APS RAW image. This dataset uses the under-development HVS-ALPIX-Eiger sensor (Alpsentek, 2024), which rearranges event and APS in a quad-Bayer pattern (a quarter photodiodes are dedicated for event, as in Fig.1). This sensor has a high resolution with $1224 \times 1632$ for events and $2248 \times 3264$ for RAW, and offers superior color and noise profiles compared to the DVS346 (Scheerlinck et al., 2019). These features make it promising for various applications (Lu et al., 2023b). We ensure the dataset diversity in two ways: photographic setting and scenes. For photographic setting, we adopt various values of aperture, focal length and exposure time. For the scene diversity, we cover 12 categories of scenes, across a wide range of color scenes, including flowers, buildings, under different weather and lighting conditions. In total, 3373 APS frames and the corresponding events are captured. A standard 24-color ColorChecker (Goto et al., 2003) is applied at certain frames as the color correction reference, as shown in Fig. 1 (c).

To generate the ground truth RGB images for the dataset, we propose a controllable ISP framework based on MATLAB (Poon & Banerjee, 2001). This ISP framework, using the ColorChecker as a prior, performs tasks such as black level calculation (Li et al., 2010), demosaicing (Hirakawa & Parks, 2006), white balance (Weng et al., 2005), denoising (Abdelhamed et al., 2018), and color correction (McElvain & Gish, 2013), resulting in high-quality RGB images with controllable errors as the reference ground truth. Since the controllable framework requires the ColorChecker information as a prior, it cannot generalize to arbitrary scenes. The color accuracy and temporal stability of this ISP are also analyzed. We categorize the existing ISP methods with RAW input into three categories and benchmark their performances on our dataset. We compare their performances across various scenarios and further conduct analysis on certain phenomena we have observed. Additionally, we

propose a simple UNet-like (Ronneberger et al., 2015) event-guided ISP neural network to fuse events with RAW images. This simple network can effectively improve the outdoor performance of ISP compared to the original UNet (Ronneberger et al., 2015). We also identify key contributions and challenges of events in the ISP process, providing a foundation and direction for future research.

## 2   RELATED WORKS

**Event-guided Imaging Datasets:** Event camera-guided imaging enhancement is an emerging field where the contribution of real datasets is crucial (Gallego et al., 2020). Currently, event cameras have made significant progress in areas such as frame interpolation (Tulyakov et al., 2021; Lu et al., 2023a; Niklaus et al., 2017; Bao et al., 2019), video super-resolution (Lu et al., 2023b; Jing et al., 2021), low-light enhancement (Liang et al., 2024; 2023), and deblurring (Xu et al., 2021; Lin et al., 2020; Jiang et al., 2020). These advancements are supported by many foundational datasets (Tulyakov et al., 2021; Scheerlinck et al., 2019; Lu et al., 2023b). For example, BS-REGB (Tulyakov et al., 2022) is a frame interpolation dataset using a beamsplitter to pair event cameras and RGB cameras. The CED (Scheerlinck et al., 2019) dataset and APLEX-VSR (Lu et al., 2023b) dataset have been used in research on event camera-guided video super-resolution. Overall, these datasets serve as the cornerstone and pioneers in research on related tasks. *However, these datasets assume that event cameras can obtain high-quality RGB images through the ISP process, an assumption that is often too idealistic.* Recognizing this, the MIPI (Yaqi et al., 2024; MIPI Challenge 2024, 2024) challenge introduced a RAW demosaic dataset for HVS in event cameras, addressing challenges like high noise and missing values in RAW from HVS. *Although this dataset is the first to focus on the RAW domain ISP process in event cameras, it lacks real event streams, thereby overlooking the potential role of events in the ISP process.* To address this gap, we propose the **first** dataset with aligned RAW and events from a new HVS, aiming at exploring the potential value and role of event data in the ISP process.

**Learning-based ISP:** Traditional ISPs (Schwartz et al., 2018) consist of long pipelines. In recent years deep learning has brought new insights to ISPs (da Silva et al., 2023a) and has achieved higher performance. These methods can be roughly categorized into three types. The first type is full pipeline replacement methods, such as PyNet (Ignatov et al., 2020b) which use CNN architectures to replace the entire ISP pipeline. The second type is stage-wise enhancement methods, like CameraNet (Liang et al., 2021) and AWNet (Dai et al., 2020), which divide the ISP pipeline into restoration and enhancement stages. The third type is image enhancement network-based methods, which utilze state-of-the-art image proessing backbone models such as UNet (Ronneberger et al., 2015) and Swin-Transformer (Liu et al., 2021) to deal with ISP tasks. Though these methods have proven effective for RAW to RGB conversion, the potential of events in this process is not explored.

**Event-guided Image/Video Enhancement:** Due to their high dynamic range and high temporal resolution (Gallego et al., 2020; Shariff et al., 2024), event cameras have garnered significant attention in the field of image/video enhancement and restoration (Gallego et al., 2020; Shariff et al., 2024), including many applications. Initially, the use of events focused primarily on single-task enhancements of RGB images or videos (Tulyakov et al., 2021; Pan et al., 2019; Lu et al., 2023b). Recently, researchers recognized image enhancement tasks are inherently coupled with various degradations interwoven (Zhang & Yu, 2022; Song et al., 2022; Yunfan et al., 2023), suggesting a trend towards using events for unified solutions in camera computational imaging for multiple tasks. *However, existing methods focus **solely** on enhancing RGB images or videos using events, overlooking the ISP pipeline, which generate RGB images from RAW images. Additionally, existing methods neglect the potential value that events could provide in the ISP process.*

## 3   DATASET COLLECTION

As the first dataset, which we call HVS-ISP Dataset, featuring paired raw-event data collected using a HVS, our aim is to facilitate research on event-guided RAW ISP. We selected the HVS-Eiger sensor developed by ALPIX (Alpsentek, 2024), which can output both APS and EVS signals that align in both time and space, as show in Fig. 2 (b). More parameter details of APS and EVS are shown in Tab. 1. Compared to the Prophesee sensor (Tulyakov et al., 2021), which can only output event signals, and the DVS346 sensor (Scheerlinck et al., 2019), which has lower resolution ($260 \times 346$)

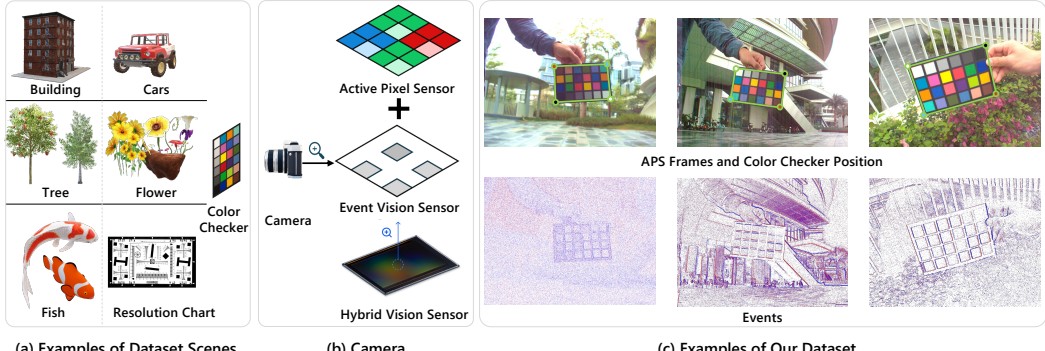

Figure 2: Overview of dataset collection. (a) illustrates the variety of scenes in the dataset, including buildings, plants, animals, and calibration boards. (b) presents a schematic of the HVS sensor, composed of a stacked active pixel sensor (APS) and an event vision sensor (EVS). (c) displays dataset samples.

Table 1: Comparison between active pixel sensor (APS) and event vision sensor (EVS) (Alpsentek, 2024) in our dataset collection. APS and EVS are stacked together to form a hybrid-vision sensor (HVS).

| Sensor | Resolution | Frame Rate | Power Consumption | Redundant Data Rate | Dynamic Range |
|---|---|---|---|---|---|
| **APS** | $2248 \times 3264$ | $10\sim60$ fps | $> 100$ mW | 10 MB/s | 60 dB |
| **EVS** | $1124 \times 1632$ | $\geq 800$ fps | $\sim10$ mW | 40-180 KB/s | $> 120$ dB |

and higher noise, our choice offers significant advantages. Hence our dataset, captured with this advanced new sensor, holds significant value for the event vision research, providing a foundation resource for advanced exploration in event-guided RAW ISP.

The collection of the dataset focuses on two main aspects: **(1)** the **diversity** of the dataset, ensuring it has broad representativeness to cover a wide range of real-world scenarios; **(2)** the inclusion of a **ColorChecker** for ISP calibration, which helps the ISP accurately restore scene colors to generate high-quality RGB frames as references.

**(1) Dataset Diversity:** In constructing our dataset, we paid particular attention to two types of diversity: camera parameter diversity and scene diversity. *Camera Parameter Diversity:* To ensure that our dataset encompasses a variety of photographic conditions, we made extensive adjustments to the camera parameters. This included aperture values ranging from $F1.0$ to $F6.0$, focal lengths extending from $8mm$ to $52mm$, and exposure times varying from $1ms$ to $100ms$. *Scene Diversity:* We focused on three key aspects to ensure comprehensive scene diversity: *Light Source Diversity:* We distinguished between indoor artificial light and outdoor natural light, with special consideration for different weather conditions. Data collection was performed under various lighting conditions, including sunny and cloudy days. *Motion Diversity:* We captured both dynamic and static videos, ensuring a mix of scenes with and without motion blur. This variety helps in testing and enhancing the performance of image processing algorithms under different motion conditions. *Material Diversity:* We included a wide array of scenes such as trees, plants, buildings, fish, dolls, and more. These scenes exhibit a broad spectrum of colors and textures, providing a rich dataset for comprehensive testing and improvement of image processing techniques.

**(2) ColorChecker as ISP Reference:** To ensure precise color correction and white balance in ISP pipeline, we utilized a standard 24-color ColorChecker (Tian et al., 2002) as critical references. At the start of each video shoot, we captured frames containing the ColorChecker and gradually removed the chart from subsequent frames. We meticulously annotated the position of the ColorChecker in each frame using the LabelMe tool (Russell et al., 2008), as shown in Fig. 2 (c). For frames without the ColorChecker, we applied previously determined ColorChecker parameters as references. This approach guarantees reliable color correction data in our dataset. Incorporating the

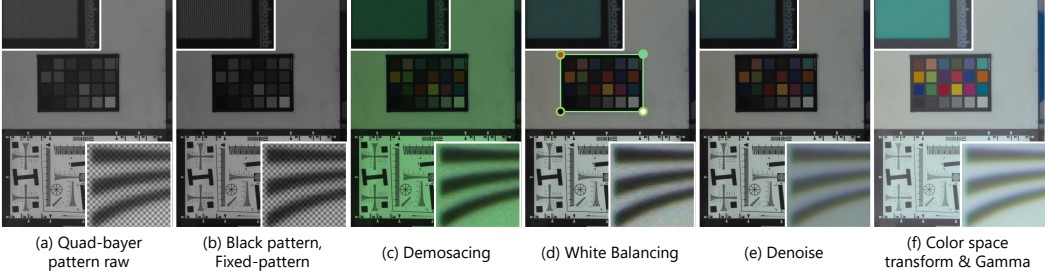

| (a) Quad-bayer pattern raw | (b) Black pattern, Fixed-pattern | (c) Demosacing | (d) White Balancing | (e) Denoise | (f) Color space transform & Gamma |

Figure 3: Flows in controllable ISP process. (a) Quad-bayer pattern raw image, which serves as the initial input. (b) Black pattern and fixed-pattern noise removal to suppress sensor-induced artifacts. (c) Demosaicing to reconstruct a rgb image from the raw data. (d) White balancing using a ColorChecker for accurate color reproduction. (e) Denoising to filter out spatial noise from the image. (f) Color space transformation and Gamma to convert the image into the desired color space for final output.

ColorChecker allows generating high-quality RGB values, enhancing color fidelity. This method ensures robustness for applications requiring accurate color restoration. Additionally, we conducted a thorough manual review of the ColorChecker annotations to validate their accuracy, further improving our dataset's reliability for ISP algorithms.

In summary, based on these two main objectives, we captured a total of 24 videos. Each video contains 80 to 140 frames, resulting in a total of 3373 APS RAW frames and their corresponding events. Additionally, the dataset includes the positions of the ColorCheckers within the APS images. We divided the dataset into training and test sets, with $3/4$ of the data used for training and $1/4$ for testing. The testing set includes 3 indoor scenes and 3 outdoor scenes to ensure sufficient diversity. *For more details on data collection and visualizations, please refer to the supplementary material.*

# 4 CONTROLLABLE ISP

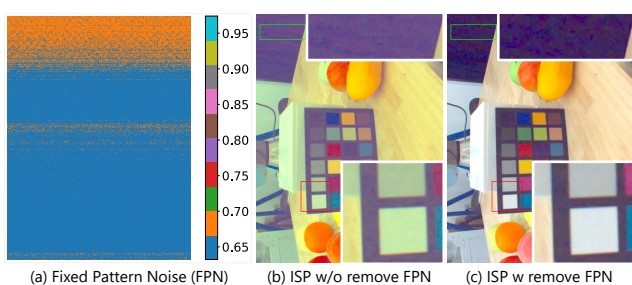

| (a) Fixed Pattern Noise (FPN) | (b) ISP w/o remove FPN | (c) ISP w remove FPN |

Figure 4: Fixed pattern noise (FPN) removal. (a) Visualizes the camera's fixed pattern noise. (b) and (c) show the RGB images without and with fixed pattern noise removal, respectively. The image in (c) demonstrates lower noise and more accurate white balance after the removal of fixed pattern noise.

The controllable ISP aims to provide module-based and analytically measurable RGB frames based on the APS RAW. With the support of the contained ColorChecker, the resulting frames have good color accuracy and low noise, serving as the reference for APS. Requirement of the ColorChecker prevents from generalizing to other arbitrary scenes. In this section, we introduce each module, followed by a quality evaluation and pros-and-cons discussion, with the hope that this ISP pipeline will be beneficial for the community.

## 4.1 CONTROLLABLE ISP PIPELINE

Fig. 3 depicts that how an image is processed via a conventional ISP pipeline, making the reference for the APS data. **(1) Black Level and Fixed Pattern Subtraction:** Taking an arbitrary unprocessed bayer raw as input, a pre-calibrated global black level value *blc* is subtracted, following by subtracting a fixed pattern vector *fpn* [1]. *blc* is the min of a raw image taken under a pure-black environment while *fpn* is a vector that records the per-row average value as the used sensor is only with horizontal fixed pattern, as shown

---
[1] *blc* and *fpn* are calibrated in a pure-dark laboratory setting. Over five frames are captured and averaged to increase the calibration accuracy.

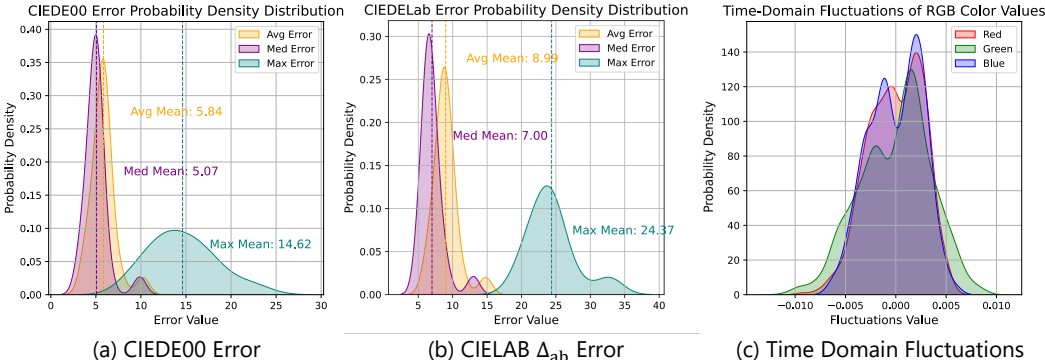

Figure 5: Color errors and fluctuations of our ISP method, computed using a ColorChecker. **(a)** CIEDE 2000 Error Probability Density Distribution: Displays CIEDE 2000 error values distribution with annotations for average (5.84), median (5.07), and maximum error means (14.62). **(b)** CIEDE Lab Error Probability Density Distribution: Shows CIEDE Lab error values distribution, indicating average (8.99), median (7.0), and maximum error means (24.37). **(c)** Time-Domain Fluctuations of RGB Color Values: Illustrates RGB color values fluctuations over time, representing temporal stability and variations in color accuracy.

in Fig. 4. **(2) Demosaicing:** Given bayer pattern, the well-adopted demosaicing method (Rainbow-Johnny-Johnny-Image-Processing-Lim, 2022) is used. The resolution is preserved while the channel number is tripled. Note that this method is still prone to generating false color in very high frequency area, as shown in Fig. 3(c). **(3) Manual White Balancing:** On a RGB image (greenish due to no white balance), we use LabelMe (Russell et al., 2008) to extract the mean colors of 24 ColorChecker patches. The $21_{st}$ patch is used as the groundtruth illumination for manual white balance Qian et al. (2017; 2019). **(4) Spatial Denoising:** We use a milestone denoising method BM3D (Dabov et al., 2009) to perform spatial denoising with the setting of $\sigma = 50$. **(5) Color Space transform:** Following Finlayson et.al. (Finlayson et al., 2015), given the retrieved ColorChecker values and the predefined oracle ColorChecker values, we optimize towards the CIEDE00 error and obtain the final color correction matrix *ccm* of the shape $(3, 3)$. A linear sRGB image is then computed from the input image I: $I_{linsrgb} = I * ccm$. **(6) Gamma:** Following sRGB standard (Anderson et al., 1996), a piecewise gamma curve is applied for brightness perception. *Due to space limitations, please refer to the supplementary material for more details and hyperparameters of controllable ISP.*

## 4.2 CONTROLLABLE ISP EVALUATION

We evaluated the controllable ISP in two main aspects: the **color accuracy** of individual images and the **temporal stability** of color recovery in continuous videos. For color accuracy, we used the CIEDE00 (Luo et al., 2001) and CIELAB $\Delta_{ab}$ (Lee & Powers, 2005) metrics to evaluate color accuracy. CIEDE00 is a widely used metric for color matching, considering the nonlinear characteristics of color differences and the human eye's sensitivity to colors, which accurately reflects human visual perception of color differences. CIELAB $\Delta_{ab}$ is a color difference metric based on the CIELAB color space (Mahy et al., 1994). Specifically, as shown in Fig. 5 (a) (b), we conducted a ColorChecker-based evaluation on 100 randomly selected samples. In CIEDE00 (Luo et al., 2001), we obtained an average value of 5.84 and a median value of 5.07; For CIELAB $\Delta_{ab}$, we obtained an average value of 8.99 and a median value of 7.00, demonstrating that our method can generally restore colors up to an accurate level. We displayed the maximum error distribution per image, showing that in CIEDE00 it is around 14, and in CIELAB $\Delta_{ab}$ around 24, affected by color filter sensitivity and photodiode layout. For temporal stability in frame estimation differences, as shown in Fig. 5 (c). We selected a 140-frame video, marking the ColorChecker in each frame. After generating colors frame by frame, we observed that differences for the 24 ColorChecker colors are all under 0.01, mostly within 0.005. This confirms our algorithm's temporal stability.

In summary, we presented a controllable ISP pipeline and analyzed its performance. However, the ISP contains numerous controllable variables and hyperparameters. We hope that future researchers will focus on optimizing these controllable aspects of the ISP to further enhance its performance.

Table 2: Comparison on Parameters, FLOPS, and Time. Top two models are highlighted in **red** and **green**.

| | Unet | PyNet | CameraNet | AWNet | PyNetCA | MW-ISPNet | InvertISP | Swin Transformer | eSL | Ev-UNet |
|---|---|---|---|---|---|---|---|---|---|---|
| **Params↓** | 16.64 | 47.55 | 25.79 | 96.07 | 29.27 | **7.22** | 92.44 | 8.87 | **0.737** | 21.51 |
| **GFLOPS↓** | 4.52 | 111.96 | 19.19 | 120.21 | 51.27 | 29.22 | **1.41** | 14.24 | 48.49 | 6.89 |
| **Time (s)↓** | **0.0100** | 0.0775 | 0.0300 | 0.2138 | 0.0308 | 0.0459 | 0.0436 | 0.0868 | 0.063 | **0.012** |

Table 3: Comparison of Methods on HVS ISP Dataset outdoor scenes. Top two models are highlighted in **red** and **green**. * refer to the results obtained by the same model with different hyperparameters.

| | 2-Out-Tree-2 | | | 3-Out-Flower-2 | | | 4-Out-Building-1 | | | Average | | |
|---|---|---|---|---|---|---|---|---|---|---|---|---|
| | PSNR↑ | SSIM↑ | $L_1$↓ | PSNR↑ | SSIM↑ | $L_1$↓ | PSNR↑ | SSIM↑ | $L_1$↓ | PSNR↑ | SSIM↑ | $L_1$↓ |
| **PyNET** | **31.70** | **0.9818** | **0.0190** | **35.12** | **0.9784** | 0.0127 | **30.60** | **0.9752** | 0.0223 | **32.47** | **0.9785** | **0.0180** |
| **PyNET*** | 27.56 | 0.9711 | 0.0310 | 32.35 | 0.9646 | 0.0175 | 28.20 | 0.9600 | 0.0311 | 29.37 | 0.9652 | 0.0265 |
| **PyNetCA** | **31.86** | **0.9788** | **0.0202** | **34.19** | **0.9773** | 0.0139 | **29.22** | **0.9725** | 0.0280 | **31.76** | **0.9762** | **0.0207** |
| **InvertISP** | 28.56 | 0.9487 | 0.0243 | 25.59 | 0.9298 | 0.0313 | 28.62 | 0.9307 | 0.0287 | 27.59 | 0.9364 | 0.0281 |
| **MV-ISPNet** | 27.05 | 0.9680 | 0.0256 | 33.61 | 0.9648 | **0.0137** | 28.62 | 0.9657 | 0.0304 | 29.76 | 0.9662 | 0.0232 |
| **CameraNet** | 11.18 | 0.2580 | 0.2289 | 12.39 | 0.2741 | 0.1899 | 10.52 | 0.2534 | 0.2609 | 11.36 | 0.2618 | 0.2266 |
| **CameraNet*** | 13.26 | 0.637 | 0.2044 | 13.59 | 0.2736 | 0.1770 | 10.06 | 0.2753 | 0.2474 | 12.30 | 0.3953 | 0.2096 |
| **AWNet** | 14.33 | 0.8836 | 0.1166 | 20.10 | 0.9316 | 0.0519 | 16.70 | 0.9390 | 0.0951 | 17.04 | 0.9180 | 0.0879 |
| **Swin-Transformer** | 25.02 | 0.9539 | 0.0308 | 29.14 | 0.9555 | 0.0231 | 21.57 | 0.9295 | 0.0523 | 25.24 | 0.9463 | 0.0354 |
| **UNet** | 21.97 | 0.9583 | 0.0393 | 29.43 | 0.9717 | 0.0208 | 22.12 | 0.9603 | 0.0460 | 24.51 | 0.9634 | 0.0354 |
| **UNet*** | 29.52 | 0.9752 | 0.0206 | 25.75 | 0.9623 | 0.0323 | 29.24 | 0.9680 | **0.0265** | 28.17 | 0.9685 | 0.0265 |
| **eSL-Net** | 25.67 | 0.9424 | 0.0342 | 19.39 | 0.9180 | 0.0576 | 24.01 | 0.9277 | 0.0502 | 23.02 | 0.9294 | 0.0473 |
| **EV-UNet** | 32.86 | 0.9795 | 0.0148 | 32.87 | 0.9698 | 0.0157 | 24.59 | 0.9600 | 0.0369 | 30.11 | 0.9698 | 0.0225 |

## 5 BENCHMARK AND DIRECTION

Based on the RGB frames obtained from the controllable ISP, we evaluate the performance of four types of ISP methods, particularly in outdoor and indoor scenarios. The experiments are conducted in the same environment and framework. Additionally, we will discuss the potential reasons behind these results and propose future research directions. **Implementation Details:** All our models were trained and tested on the same machine with a single A40 GPU with 48GB of GPU memory. We used PyTorch (Paszke et al., 2017) for all experiments, applying random cropping and rotation for data augmentation. The training batch size was 1, with each patch sized at $1024 \times 1024$. The learning rate was 0.0001, and all models were trained for 50 epochs. **Evaluation Metrics:** We evaluate model performances in two aspects: resource consumption, including parameters in millions ($M$), GFLOPS, and average inference time ($s$); and image reconstruction for indoor and outdoor scenes, measured by PSNR (Hore & Ziou, 2010), SSIM (Brunet et al., 2011), and $L_1$ distance.

### 5.1 ISP BENCHMARK METHODS

Inspired by the prior ISP survey study (da Silva et al., 2023b), we categorize learning-based ISP models into three classes: full pipeline, stage-wise, image enhancement network-based. We selected two to four open-source models from each category for training and evaluation on our dataset. Furthermore, we put forward another new category of event fusion method, and since there is no prior research to refer to, we design a simple event-guided ISP neural network to test on our dataset. *For more details on ISP methods, please refer to the supplementary material.*

**Full Pipeline ISP:** These models utilize CNN architectures to integrate traditional ISP processes into an end-to-end conversion from RAW to RGB images. Notable models in this category include PyNet (Ignatov et al., 2020b), PyNetCA (Kim et al., 2020), InvertISP (Xing et al., 2021), and MV-ISPNet (Ignatov et al., 2020a).

**Stage-wise ISP:** They employ multiple specialized modules to handle different ISP tasks, either sequentially or in parallel, to produce the final image. In our benchmark, we selected CameraNet (Liang et al., 2021) and AWNet (Dai et al., 2020) for their distinct approaches. Note that due to the unavailability of a PyTorch version of CameraNet (Liang et al., 2021), we experimented on a converted version. The modules in the original AWNet (Dai et al., 2020) are trained independently, however in our experiment we trained them end-to-end.

**Image Enhancement Network-Based ISP:** There have been numbers of high performance backbone models for image enhancement in image enhancement tasks like deblurring (Zhang et al.,

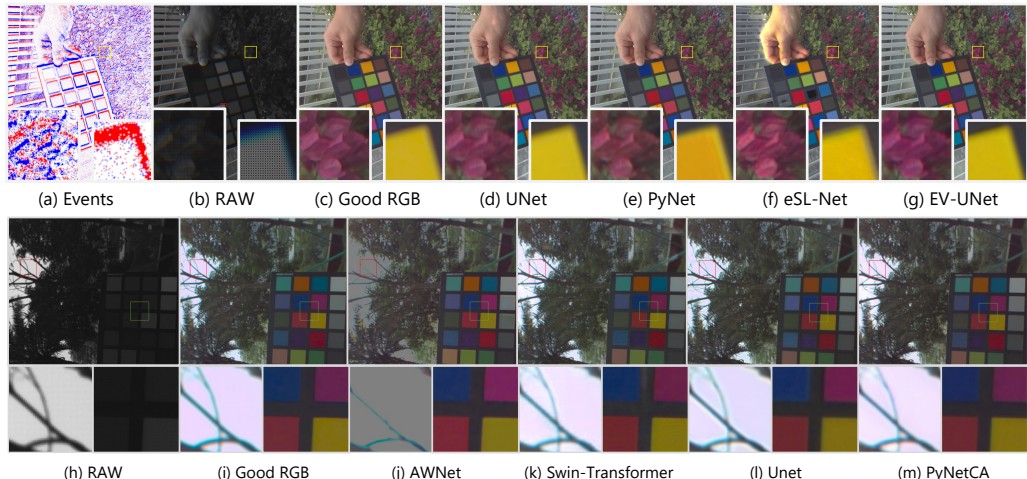

(a) Events (b) RAW (c) Good RGB (d) UNet (e) PyNet (f) eSL-Net (g) EV-UNet

(h) RAW (i) Good RGB (j) AWNet (k) Swin-Transformer (l) Unet (m) PyNetCA

Figure 6: Visualization results of different methods on HVS-ISP Dataset outdoor scenes.

Table 4: Comparison of Methods on HVS ISP Dataset indoor scenes. * refer to the results obtained by the same model with different hyperparameters.

| Methods | 1-In-Fruit-2 PSNR↑ | SSIM↑ | $L_1$↓ | 3-In-ColChecker-40 PSNR↑ | SSIM↑ | $L_1$↓ | 4-In-RLChart-10 PSNR↑ | SSIM↑ | $L_1$↓ | Average PSNR↑ | SSIM↑ | $L_1$↓ |
|---|---|---|---|---|---|---|---|---|---|---|---|---|
| PyNET | 13.09 | 0.7970 | 0.2182 | 11.38 | 0.7922 | 0.2489 | 11.42 | 0.7100 | 0.2563 | 11.97 | 0.7664 | 0.2412 |
| PyNET* | 14.46 | 0.8068 | 0.2008 | 24.02 | 0.9550 | 0.0497 | 13.58 | 0.7694 | 0.1978 | 17.36 | 0.8437 | 0.1494 |
| PyNetCA | 18.13 | 0.8843 | 0.1253 | 29.53 | 0.9723 | 0.0246 | **35.51** | **0.9727** | **0.0121** | 27.72 | 0.9431 | 0.0540 |
| InvertISP | 25.83 | 0.9098 | 0.0346 | 28.33 | 0.9500 | **0.0235** | 30.65 | 0.9578 | 0.0183 | 28.27 | 0.9392 | 0.0254 |
| MV-ISPNet | **31.91** | **0.9594** | **0.0185** | 29.56 | **0.9729** | 0.0265 | 31.88 | 0.9670 | 0.0170 | **31.12** | 0.9664 | **0.0207** |
| CameraNet | 13.06 | 0.2660 | 0.1947 | 13.58 | 0.2722 | 0.1836 | 12.47 | 0.2391 | 0.2257 | 13.04 | 0.2591 | 0.2013 |
| CameraNet* | 14.18 | 0.290 | 0.1630 | 10.60 | 0.2667 | 0.2545 | 13.26 | 0.2636 | 0.2044 | 12.68 | 0.2672 | 0.2073 |
| AWNet | 17.95 | 0.8665 | 0.1302 | **32.17** | **0.9807** | **0.0184** | 30.98 | 0.9596 | 0.0215 | 27.03 | 0.9356 | 0.0567 |
| Swin-Transformer | 25.73 | 0.9397 | 0.0301 | 25.50 | 0.9561 | 0.0359 | 26.18 | 0.9486 | 0.0252 | 25.80 | 0.9481 | 0.0304 |
| UNet | 17.62 | 0.9161 | 0.0747 | 13.96 | 0.8828 | 0.1454 | 15.53 | 0.8750 | 0.1170 | 15.70 | 0.8913 | 0.1124 |
| UNet* | **32.52** | **0.9659** | **0.0161** | 29.04 | 0.9740 | 0.0257 | **33.72** | **0.9716** | **0.0146** | **31.76** | **0.9705** | **0.0188** |
| eSL | 27.09 | 0.9428 | 0.0331 | 24.79 | 0.9548 | 0.0434 | 26.52 | 0.9415 | 0.0379 | 26.13 | 0.9464 | 0.0381 |
| EV-UNet | 14.16 | 0.8706 | 0.1533 | 31.64 | 0.9779 | 0.0214 | 32.33 | 0.9678 | 0.0173 | 26.04 | 0.9388 | 0.0640 |

2022) and super-resolution (Chen et al., 2022). Though not initially designed for ISPs, minor modifications can adapt these models for ISP tasks. For our benchmark, we selected UNet (Ronneberger et al., 2015) and Swin-Transformer (Liu et al., 2021; Lu et al., 2024).

**Event Fusion Method:** As the first research on event-guided ISP, we have no prior research for reference. Therefore, we selected eSL-Net (Wang et al., 2020a), an event-based backbone network used in various tasks (Lu et al., 2023b). Additionally, we merged events as voxel-grid (Liu et al., 2023) with UNet's encoder as EV-UNet to verify events effectiveness and challenges.

## 5.2 COMPARATIVE EXPERIMENTS AND VISUALIZATION ANALYSIS

**Computational Performance:** In Tab. 2, InvertISP (Xing et al., 2021) excels in computational efficiency with 1.41 GFLOPS, significantly lower than the over 100 GFLOPS of AWNet (Dai et al., 2020) and PyNet (Kim et al., 2020), which is suitable for limited computing resources. UNet surpasses CameraNet (Liang et al., 2021) in processing speed with a response time of 0.01 s, preferable for real-time performance. Overall, UNet demonstrates balanced performance with low GFLOPS and the fastest processing speed, due to its straightforward design.

**Outdoor Performance:** Tab. 3 shows the superior performance of PyNet across three outdoor backgrounds. PyNet (Kim et al., 2020) achieves the best PSNR (Hore & Ziou, 2010), SSIM (Brunet et al., 2011), and $L_1$ with an overall average PSNR (Hore & Ziou, 2010) of 32.47, significantly higher than other models. Specifically, EV-UNet shows significant improvement in outdoor scenes with UNet after incorporating events gain, increasing from 28.17 to 30.11. In contrast, the commonly used event-based method eSL-Net performs poorly with a PSNR (Hore & Ziou, 2010) of only 23. This poor performance mainly results from the *limited receptive field* of eSL, which is insufficient for

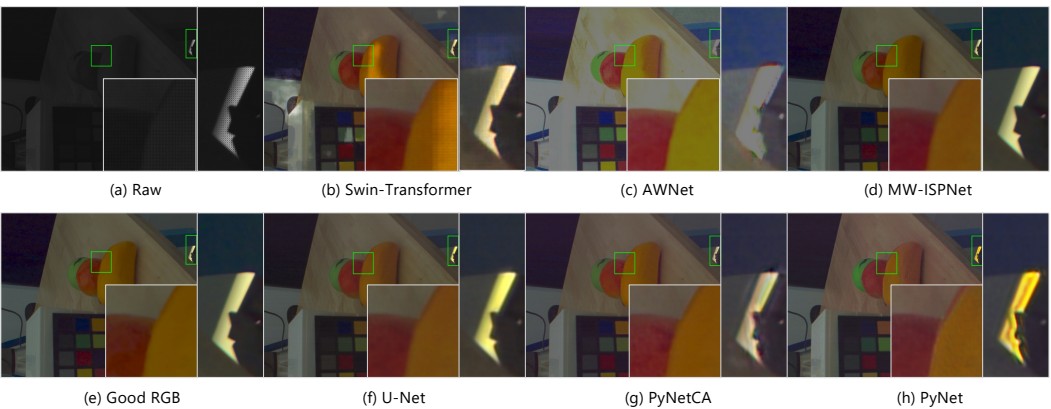

Figure 7: Visualizations on HVS-ISP Dataset indoor scenes.

estimating the **global illumination** information, and thus failing to achieve consistent global illumination enhancement. We further discuss on this issue in Sec. 5.3. we also visualize the results in Fig. 6. PyNet has achieved the highest PSNR (Hore & Ziou, 2010) but exhibits edge artifacts, this is likely due to the overfitting of the model. In outdoor scenes, event-enhanced outputs of EV-UNet show good global consistency. Fig. 6 shows that AWNet (Dai et al., 2020) struggles with fine texture restoration, explaining its inferior performance to other methods.

**Indoor Performance:** Tab. 4 shows that UNet* excels in indoor environments, especially when handling multiple colored fruits and scenes with complex lighting and details. The output of AWNet (Dai et al., 2020) has overall excessive brightness, as illustrated in Fig. 7, explaining its low PSNR values. PyNet exhibits noticeable artifacts, consistent with the good RGB edge but with significantly different brightness, likely due to the ill-posed nature of brightness recovery in the ISP process, resulting in its poor indoor performance. Event-fusion methods perform poorly indoors, primarily due to flickering light sources that complicate event characteristics. For more analysis about these issues, please refer to Sec. 5.3.

**Summary:** These sections show that the performance of numerous ISP methods on HVS sensor datasets varies significantly across different scenes. For instance, PyNet and AWNet (Dai et al., 2020) exhibit great variability between indoor and outdoor environments, underscoring that learning-based ISP methods are highly scene-dependent. This highlights the necessity for future work to analyze different scenes individually to fully understand the performance of a network. Furthermore, adding events to UNet significantly improves performance in outdoor scenarios but not indoors, mainly due to the flickering indoor lighting. Addressing this issue remains a crucial challenge for future research.

## 5.3 DISCUSSION AND FUTURE DIRECTION

Through the comprehensive and objective evaluation of various models on our dataset, we have also observed a number of findings that can bring insights for future work.

**Significant Indoor-Outdoor Performance Gap on PyNet and AWNet (Dai et al., 2020):** We observed a significant indoor-outdoor performance gap on PyNet (Ignatov et al., 2020b) and AWNet (Dai et al., 2020). PyNet performs better in outdoor scenes than indoor, ranking the top of all models, while AWNet (Dai et al., 2020) shows quite the opposite behavior. Generally, outdoor scenes have more dynamic and varied lighting compared to indoor environments, which are difficult for models to learn. The original AWNet (Dai et al., 2020) is designed to be trained in a multi-stage manner with different loss functions. Therefore it might have fallen into sub-optima when trained end-to-end in our experiment, resulting in the poor performance in modeling the harder outdoor scenes.

**Local Brightness Artifacts:** Artifacts occur when the brightness in certain image areas significantly deviates from the overall luminance (see Fig. 7). We investigated this by examining the relationship between a brightness of a pixel and the RAW data within its $5 \times 5$ vicin-

ity. We treat the neighboring RAW data as a 25-dimensional vector, and apply t-SNE to project it onto a 2D plane, recording the $(x, y)$ coordinates. We then converted the RGB values of the pixel to YUV, recording the Y (brightness) as the $z$ coordinate, as shown in Fig. 8.

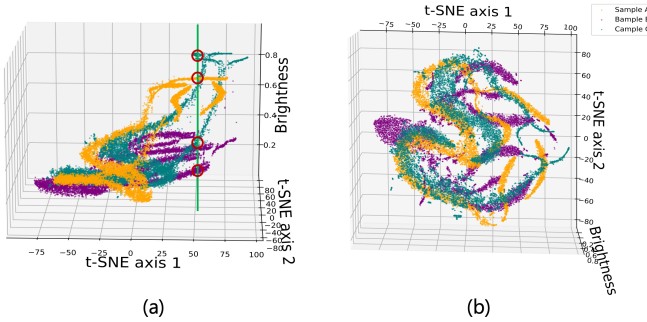

By plotting pixels from three random images in 3D (Fig. 8), we show that pixel brightness and neighboring RAW data have a non-injective relationship. Multiple brightness levels can emerge from the same RAW data, indicating that **global information**, not just local RAW value, is essential for accurately determining pixel brightness to avoid local artifacts.

Figure 8: The ill-posedness of brightness estimation in the ISP process. We visualized the $5 \times 5$ region in the RAW image and the brightness of corresponding pixel in the color image at the center of this region. The results show that the same RAW region corresponds to different brightness levels in different images.

**Event Gains:** The integration of events in our dataset significantly enhances performance in outdoor scenes when comparing EV-UNET with UNet, primarily due to the additional motion information and dynamic range provided by the events. However, simple fusion does not fully exploit these characteristics, highlighting the need for more sophisticated designs in future research. Conversely, performance decreases in indoor scenes, primarily due to the flickering of artificial light sources.

**Flickering Artificial Lighting:** Under certain indoor scenarios, some artificial light source (Xu et al., 2023), *e.g.*LEDs, flicker because of the alternating current frequency. Given that the event frame rate of the sensor significantly exceeds the usual AC frequency (50 or 60 Hz), the flickering lighting introduces considerable fluctuations in the event data over time. The distributions and features of events in these conditions are completely different from that in the natural lighting conditions, and could result in the model's failure in restoring the images from RAW data.

## 6 CONCLUSION

In this work, we present the first events-RAW paired dataset for event-guided ISP research. The dataset consists of 3373 high quality high resolution RAW images and corresponding **pixel-level aligned** events. Subsequently, good RGB frames are generated by a controllable ISP pipeline we proposed. A comprehensive evaluation and analysis of existing learnable ISPs and a simple event-guided ISP method are conducted on our dataset. Based on this analysis, we summarize some key points and challenges for event-guided ISP. We wish to emphasize the potential of event data in ISP processes again. Event cameras have a high dynamic range and high temporal resolution, which surpass the limits of human vision systems. In terms of dynamic range and temporal sampling, the information captured by event sensor is somehow a superset of that of human eye. Therefore, generating images perceptible to human vision is a matter of downward compatibility. **Limitations:** Firstly, the scale of our dataset is relatively small, because the HVS sensor we use is still in the prototype stage and the associated hardware is cumbersome and exhibits low stability, which has raised the cost in data collection and thus a limited size dataset. And yet we are committed to expanding the dataset with more diverse real-world scenarios in future research. Secondly, our dataset has not thoroughly addressed the issue of flickering in artificial lighting caused by alternating current, especially in indoor scenarios. The flickering considerably impairs the performance of our method and further research should pay attention to this problem.

**Acknowledgements:** This work was supported in part by the National Key R&D Program of China (Grant No.2023YFF0725001),in part by the National Natural Science Foundation of China (Grant No.92370204), in part by the Guangdong Basic and Applied Basic Research Foundation (Grant No.2023B1515120057), in part by Guangzhou-HKUST(GZ) Joint Funding Program (Grant No.2023A03J0008), Education Bureau of Guangzhou Municipality.

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
