# OpenReview forum: "RGB-Event ISP: The Dataset and Benchmark"
_ICLR.cc/2025/Conference — ICLR 2025 Poster_

### Official Review · Reviewer_vrds · 2024-10-16

**Soundness:** 2
**Presentation:** 3
**Contribution:** 2
**Rating:** 6
**Confidence:** 3

**Summary:**

The paper presents a new event-RAW paired dataset collected with a novel but private sensor that records pixel-level aligned events and RAW images. 3373 RAW images with paired events spanning 24 scenes are captured. A convential ISP pipeline is proposed to generate RGB references and learnable ISP methods are used to train and evaluate on the dataset.

**Strengths:**

1. Event-guided ISP seems like a novel and interesting idea.
2. A novel sensor is designed to capture the datasets, although it's private and confidential.
3. The analysis and results looks good.

**Weaknesses:**

1. The proposed is rather small-scaled. There are 24 videos captured in total, with 80 to 140 frames for each video. Considering a FPS of 60, it's just 1-2 seconds. It would be better called an image dataset instead of a video dataset.
2. I assume the dataset would be open-sourced? Although the paper does not explicitly state that.
3. As the authors stated, the prototype is cumbersome with low stability and also private. This weakens the reproducibility of the work.
4. The background of the dataset and issues need further clarifications. It is not clear to me at the moment.

**Questions:**

Please explain what is 'event' (with examples) and why event camera is important in addition to RGB camera, maybe in the introduction/related work section. This is important for readers unfamiliar with this topic.

---

> ### Author Response · Authors · 2024-11-19
>
> Dear Reviewer, vrds:
>
> Thank you for your valuable suggestions.
> We have carefully revised the paper to address your concerns to the fullest extent possible. Specifically, we have added Section A and Section B in the supplementary materials (pages 16–20) to provide detailed explanations about the imaging process, underlying principles, and dataset scale. These additions aim to clarify and enhance the understanding of the key aspects you highlighted.
>
> Below are specific answers to your questions.
>
> ### Q.1 Concern about Dataset Size for ISP tasks:
> Thank you for your suggestion and concern. We have answered the question about the scale of the dataset in Summary of Official Reviews (2/2). We hope that this will resolve your doubts.
>
> ### Q.2 Open-Sourcing the Dataset
> We promise that the dataset, benchmark code and pre-train model will be open sourced upon the acceptance of this paper. This will allow the broader community to access, evaluate, and build upon our work.
> A clear statement regarding dataset release has been added to the paper.
>
> ### Q.3 Concern about Reproducibility
> Thank you for your concern regarding the reproducibility of our experiments with the sensor. We appreciate this opportunity to provide further clarification.
> We would like to emphasize that the sensor is expected to be commercialized and made publicly available in the near future.
>
> Our research serves as an early-stage exploration of its potential applications, aiming to uncover the benefits and challenges of integrating such technology into ISP tasks. We hope that this work can inspire further advancements in this field and help guide the community's adoption and innovation around this promising technology.
>
> ### Q.4 & Q.5 More Background of Dataset & More Explain about Events
> Thank you for your suggestion. We agree that clarifying the dataset’s background and unique characteristics will greatly benefit readers.
> In the appendix, we have added section A to explain the imaging principles, features, and unique advantages of this sensor. Additionally, we have included new examples to highlight its performance in challenging scenarios, such as high-speed motion and low-light conditions, demonstrating the benefits of this hybrid sensor.
>
> Best

---

> ### Author Response · Authors · 2024-11-25
> **Looking forward to further discussions with reviewer vrds.**
>
> Thank you for your thoughtful review. In the revision paper, we have incorporated additional background explanations to address your concerns. We hope these revisions provide clarity and resolve the issues you raised. We look forward to engaging in further discussions with you.

---

> > ### Comment · Reviewer_vrds · 2024-11-27
> >
> > Thanks for your hard work. My concerns are addressed and I increased my rating to 6.

---

> > > ### Author Response · Authors · 2024-11-27
> > > **Grateful for Your Feedback and Support**
> > >
> > > Thank you for your kind and encouraging response! We are delighted to hear that our revisions have addressed your concerns and clarified the points you raised. Your decision to increase your score is deeply motivating for us and inspires us to continue striving for excellence in RGB Event ISP.
> > >
> > > We deeply appreciate the time and effort you have dedicated to reviewing our paper and providing constructive feedback. Your insights have contributed to improving the quality of this research. Once again, thank you for your time, effort, and helpful feedback!

---

### Official Review · Reviewer_fVak · 2024-10-29

**Soundness:** 3
**Presentation:** 3
**Contribution:** 2
**Rating:** 8
**Confidence:** 4

**Summary:**

This paper introduces the RGB-Event ISP dataset and benchmark, a novel paired dataset that combines RAW and event data from hybrid vision sensors to support research in event-guided image signal processing (ISP). The authors design a customizable ISP pipeline, allowing for benchmarking ISP methods and exploring the benefits of integrating event data in RAW-level ISP tasks. The experiments demonstrate the effectiveness of several methods in both outdoor and indoor scenes, and the dataset offers unique insights into ISP improvements through event data.

**Strengths:**

1. The dataset fills an existing gap by providing paired RAW and event data, which is the first of its kind specifically designed for ISP tasks.
2. The proposed ISP pipeline is flexible, covering key ISP stages like black level adjustment, demosaicing, and color correction, making it versatile for testing various ISP algorithms.
3. The experimental evaluation is thorough, comparing several state-of-the-art ISP methods and highlighting the advantages of using event data, particularly in dynamic outdoor scenes.
4. The paper also addresses its limitations and suggests potential solutions. Specifically, it provides practical insights into the challenges of integrating event data with ISP, including issues like artifacts under artificial indoor lighting.

**Weaknesses:**

1. The dataset size is relatively small (3373 images), which might limit the generalizability of models trained on it. Additional data could enhance the applicability of the findings.
2. The theoretical justification for how event data contributes to each ISP stage could be more developed. For instance, further details on how event data specifically enhances tasks like white balancing and noise reduction would make the contribution clearer.
3. The writing lacks clarity, making the paper difficult to understand for readers not deeply familiar with the field. Enhancing the language clarity and providing more structured explanations would significantly improve the paper's readability and accessibility.

**Questions:**

1. In Section 3.1, it would be helpful to elaborate on the role of event data in each ISP stage (e.g., how it improves demosaicing or color correction compared to using RAW data alone).
2. There are some spelling errors in the paper. For instance, both "convential ISP pipeline" and "conventional ISP pipeline" appear, where "convential" is incorrect; it should be "conventional ISP pipeline." Additionally, in Figure 5c, the y-axis should be labeled as "probability density," but the word "density" seems to have been omitted.
3. My concern is that some comparative algorithms have very low PSNR scores. From the varying performance of PyNET and UNet under different hyperparameters, I believe it would also be beneficial to test CameraNet with different hyperparameters. This would allow us to select a high-performance configuration as a baseline, ensuring a fairer and more comprehensive comparison.
4. I have a question regarding the alignment between APS (Active Pixel Sensor) and EVS (Event-based Vision Sensor) data. As mentioned in the paper, there is a significant difference in their frame rates. How is the event data aligned with the RAW data given this disparity in FPS?

---

> ### Author Response · Authors · 2024-11-19
>
> Dear Reviewer fVak:
>
> Thank you for your thoughtful feedback and valuable suggestions. We greatly appreciate the time and effort you have taken to review our work. Below, we provide detailed responses to your concerns and describe how we have addressed them in the revised manuscript.
>
> ### Q.1 Concern about Dataset Size
> Thank you for pointing this out.
> This concern has been discussed in detail in the section "Summary and Answers of Official Reviews (2/2)." Please refer to this section for our comprehensive response.
>
>
> ### Q.2 Elaboration on the Role of Event Data in Each ISP Stage
> Thank you for raising this important point. To clarify, in the Section CONTROLLABLE ISP stage, we used a ColorChecker-based ISP pipeline to generate reference images as ground truth. In this stage, tasks such as demosaicing and color correction are performed using traditional computational methods. The evaluarion results are demonstrated in Figure 5 of the main paper. This process does not involve the use of event data.
>
> Inspired by your suggestion, we have included additional analysis in the supplementary materials to explore how event characteristics could assist in improving RAW-level ISP tasks when integrated into deep learning frameworks. We believe this will provide valuable insights for future research directions and further demonstrate the potential of event-guided ISP.
>
> ### Q.3 Refine Writing
> Thank you for highlighting the need for improved clarity in writing. We have thoroughly reviewed the manuscript to correct spelling errors and ensure precise language throughout. Your feedback has helped us significantly improve the readability and robustness of the paper.
>
> ### Q.4 Hyperparameters of Comparative Algorithms
> Thank you for the suggestion. We have conducted additional experiments with CameraNet using optimized hyperparameters. These results, included in the revised manuscript, provide a more comprehensive and fair comparison of baseline methods.
>
> ### Q.5 Alignment of EVS and APS
> We appreciate your question regarding the alignment between EVS (Event-based Vision Sensor) and APS (Active Pixel Sensor) data. In the appendix (Section A, Pages 16–19), we provide a detailed explanation of the temporal alignment process. Specifically, the EVS and APS sensors are equipped with unified timestamps, enabling precise alignment at the pixel level. This ensures that event data is synchronized with the rolling shutter frames of the APS output, allowing for accurate integration in hybrid sensor tasks.
>
> Once again, we thank you for your constructive feedback. Your comments have not only helped us address key concerns but also inspired us to refine our work and present a more robust contribution to the field. We hope our responses and revisions address your concerns satisfactorily.
>
> Best

---

> > ### Comment · Reviewer_fVak · 2024-11-27
> > **Acknowledging Clarifications and Adjusting Assessment**
> >
> > Thank you for addressing my questions. I feel that I have gained a deeper understanding of the paper, so I have increased my score and confidence accordingly.

---

> > > ### Author Response · Authors · 2024-11-27
> > > **Grateful for Your Encouraging Support**
> > >
> > > Thank you for your thoughtful and constructive feedback. We are very pleased to hear that our clarifications have addressed your concerns and helped improve your understanding of the paper. We are truly grateful for your decision to increase your score, as it greatly encourages and motivates us.
> > > Your valuable suggestions have played an important role in enhancing the quality of our research, and we sincerely appreciate your support. Once again, thank you for your time, effort, and helpful feedback.

---

> ### Author Response · Authors · 2024-11-25
> **Looking forward to further discussions with reviewer fVak.**
>
> Thanks for your time and attention. In the revision paper, we have addressed the dataset scale, the data alignment methodology, and provided additional background information. We hope you have had the opportunity to review the revised version. We are eager to discuss any further questions or feedback you may have.

---

### Official Review · Reviewer_AvAS · 2024-11-04

**Soundness:** 3
**Presentation:** 3
**Contribution:** 2
**Rating:** 6
**Confidence:** 4

**Summary:**

The authors introduce the first events-RAW paired dataset specifically designed for event-guided image signal processing (ISP) research. This dataset comprises 3,373 high-quality, high-resolution RAW images alongside corresponding pixel-level aligned events. Using a controllable ISP pipeline developed by the authors, high-quality RGB frames are generated. A thorough evaluation and analysis of existing learnable ISPs, as well as a straightforward event-guided ISP method, are performed on this dataset. From this analysis, the authors highlight several key points and challenges associated with event-guided ISP.

**Strengths:**

1. The relevant background knowledge of this paper is clearly explained.

2. The topic of RGB-event ISP is very interesting topic and meanful for future camera.

3. The writing is very well and easy to understand.

**Weaknesses:**

1. I greatly appreciate the effort to create a dataset for RGB-Event ISP, which opens up opportunities for event-assisted RGB ISP tasks. However, the dataset's scale is quite limited, with only 3,373 samples, which may not be sufficient to support data-driven learning methods. This raises concerns about the dataset's ability to serve as a professional, standardized, and challenging benchmark. If this is primarily a workload issue, could the authors consider generating some simulated datasets? I would appreciate an explanation regarding this.

2. The authors use images generated from a controllable ISP framework based on MATLAB as ground truth. While they provide extensive explanations for this approach, it is difficult to trust software-generated images as ground truth, especially for a professional dataset. This practice differs significantly from the ground truth methods used by existing sensor or smartphone manufacturers for their ISPs.

3. The integration of events into traditional ISP theoretically brings certain advantages, and the authors should elaborate on these benefits. Additionally, to demonstrate the advantages of event cameras, the authors should showcase scenes, particularly under high-speed motion or extreme lighting conditions, that highlight the potential benefits of using event-based approaches.

**Questions:**

Please see the weaknesses. I have assigned a preliminary score based on the initial manuscript and will consider adjustments depending on the authors' responses and feedback from other reviewers.

---

> ### Author Response · Authors · 2024-11-19
>
> Dear Reviewer AvAS,
>
> Thank you for taking the time to review our work and for providing thoughtful and constructive feedback. We are grateful for your recognition of the following strengths in our paper:
> (1) Background: Clearly explained and accessible to readers.
> (2) Relevance: RGB-event ISP is an interesting and meaningful topic for future cameras.
> (3) Writing: Well-structured and easy to understand.
> We have carefully reviewed your comments and suggestions. Below, we provide detailed responses to your specific concerns.
>
> ### Q.1 Dataset Scale and Simulated Data Generation
> Thank you for raising this concern. Compared to the most recent similar datasets, our dataset is four times larger in scale. More importantly, our dataset is based on real-world data and includes event outputs, which are critical for hybrid sensor ISP tasks. For a more detailed response, please refer to "Summary and Answers of Official Reviews (2/2)."
>
> Additionally, we appreciate your suggestion about expanding the dataset. We are committed to the long-term maintenance and future expansion of this dataset to include more scenes and conditions. We also believe the current scale is already sufficient to support training and testing models effectively.
>
>
> ### Q.2 Trustworthiness of Ground Truth Generated by MATLAB
> We understand your concerns regarding the use of MATLAB-based tools for generating ground truth. To clarify:
>
> - The MATLAB-based controllable ISP framework is a widely recognized tool in academic research for generating high-quality ground truth images. This framework allows us to systematically control key ISP stages, such as demosaicing [a], denoise [b] and color correction [c].
> In traditional ISP tasks, this approach is a standard and reliable method in the academic community [d].
>
> - To ensure transparency, we have included detailed descriptions of the framework in the manuscript and have conducted quantitative evaluations to validate the accuracy and consistency of the generated ground truth images. These are illustrated in Figure 5 of the main paper and Figure 13 of the Appendix.
>
> Additionally, in practical scenarios, consumer cameras imaging often lacks precise color references.
> By including a ColorChecker in every scene and using it to generate accurate color correction matrices, our approach ensures greater color accuracy than methods without such reference data.
>
> ### Q.3 Integration of Events into Traditional ISP
> Thank you for this valuable suggestion. We have expanded the discussion of the advantages of integrating events into traditional ISP tasks in the revised manuscript. Specifically:
>
> Practical Examples: In Section A, Figure 10 of the supplementary materials, we present examples in low-light and fast-motion scenarios. These demonstrate how events provide high temporal resolution and wide dynamic range, highlighting their advantages over RGB-based approaches.
>
> Theoretical Analysis: Based on your suggestion, we have added a new section in the supplementary materials analyzing why event data is effective for ISP tasks, including demosaicing and color correction. This analysis provides a theoretical foundation for integrating events into ISP.
>
> Once again, thank you for your constructive feedback. Your comments have significantly improved the clarity and impact of our work. We hope our responses adequately address your concerns and demonstrate the robustness of our contributions.
>
> Best
>
> ## Reference:
> - [a] Malvar, H.S., L. He, and R. Cutler, High quality linear interpolation for demosaicing of Bayer-patterned color images. ICASPP, Volume 34, Issue 11, pp. 2274-2282, May 2004.
> - [b] Metzler, Christopher A., Arian Maleki, and Richard G. Baraniuk. "BM3D-AMP: A new image recovery algorithm based on BM3D denoising." 2015 IEEE international conference on image processing (ICIP). IEEE, 2015.
> - [c] Westland, Stephen, Caterina Ripamonti, and Vien Cheung. Computational colour science using MATLAB. John Wiley & Sons, 2012.
> - [d] Sumner, Rob. "Processing raw images in matlab." Department of Electrical Engineering, University of California Sata Cruz 2 (2014).

---

> > ### Comment · Reviewer_AvAS · 2024-11-27
> >
> > Thank you for your thoughtful response. The small size of the dataset is one of the reasons why this paper may not stand out as much, especially since it's focused on dataset and benchmark development. However, considering this is my first attempt at such work, I’ve decided to maintain the original positive score. Wishing you the best of luck!

---

> > > ### Author Response · Authors · 2024-11-27
> > > **Grateful Acknowledgment and Commitment to Advancing RGB-Event ISP**
> > >
> > > Thank you for your kind and encouraging feedback!
> > > We deeply appreciate your understanding and recognition of our efforts in advancing this work. Your acknowledgment inspires us to continue contribute to the development of the RGB-Event ISP field.
> > >
> > > Thank you again for maintaining your positive score, and we sincerely wish you all the best as well!

---

> ### Author Response · Authors · 2024-11-25
> **Looking forward to further discussions with reviewer AvAS.**
>
> Thank you for your appreciation of our work. In the revision paper, we have discussed the dataset scale in detail and provided additional examples. Additionally, we have demonstrated the advantages of this camera in motion and low-light scenarios. We hope the revision paper has addressed your concerns and captured your interest. We look forward to engaging with you further.

---

### Official Review · Reviewer_qDHH · 2024-11-04

**Soundness:** 2
**Presentation:** 3
**Contribution:** 3
**Rating:** 5
**Confidence:** 2

**Summary:**

This paper presents a novel dataset tailored for event-guided image signal processing (ISP), featuring 3,373 high-resolution RAW images paired with corresponding event data across various scenes, exposure modes, and lenses. This dataset aims to advance RGB-Event ISP research by enabling the development of methods that directly incorporate event information within the ISP pipeline. Besides, they introduce an event-guided ISP neural network as a baseline, fusing events with RAW data to optimize ISP tasks. Then, various ISP methods are evaluated on this dataset, establishing a foundation for future RGB-Event ISP advancements. Note that this work generated high-quality RGB images as ground truth by using a ColorChecker.

**Strengths:**

1.	Dataset: This is the first dataset with aligned (in both time and spatial space) RAW and events from a new HVS equipment, which may provide some avenues for developing new ISP algorithms.
2.	Benchmark Task: This work evaluates the proposed dataset by testing various ISP baseline methods along with an event-guided ISP approach. Some conclusions and insights can be drawn from these experiments.

**Weaknesses:**

1.	Scale of the dataset: From my understanding, a robust dataset should have both scale and diversity. Although the authors mention that the dataset is relatively small due to the low stability of the HVS sensor, I am not completely convinced by it, and I still believe it would be beneficial to expand the dataset further to increase its richness and variety. Additionally, the authors could include suggestions in the future work section on strategies for scaling up the dataset.
2.	Reproducibility and Usage of the dataset: Given that this is a dataset/benchmark paper, I strongly encourage the authors to release the data as soon as possible, even during the review stage. Early access is crucial for the community to begin evaluating and utilizing the dataset.
3.	Benchmark Task: This dataset is tailored for RAW-Event ISP tasks. So, I wonder if we should focus more on designing various Event-guided ISP baselines. In the current setup, most baselines do not utilize the event data, raising concerns about whether these limited event-guided baselines effectively demonstrate the dataset’s potential. I believe increasing the variety of event-driven baselines could better validate the usefulness of this dataset and highlight its unique contributions to event-guided ISP research.

**Questions:**

My questions are highly overlapped with the weakness as follows:
1.	Dataset Scale and Diversity: Could you provide more insights into the limitations that prevented a larger dataset collection, and provide more insights about how to scale up the data?
2.	Early Data Release: Is there a timeline or plan for public release of the dataset, especially considering its importance for community validation and use?
3.	Event-Guided ISP Baseline Evaluation: Given that the dataset is designed for RAW-Event ISP tasks, consider developing additional baseline methods that explicitly incorporate event data to better assess the dataset's strengths. For instance, could you explore a wider variety of event-guided architectures or fusion methods to better showcase the potential of the event data in ISP?

---

> ### Author Response · Authors · 2024-11-19
>
> Dear Reviewer qDHH,
>
> Thank you for your thoughtful review and for highlighting the key contributions of our work. We greatly appreciate your recognition of the following strengths.
> (1) Dataset: The first of its kind with aligned RAW and event data from a hybrid vision sensor, providing new opportunities for ISP algorithm development.
> (2) Benchmark Task: Evaluating the dataset with various ISP methods and deriving valuable insights for event-guided ISP research.
> Below, we address each of your concerns in detail.
>
> ### Q.1 Dataset Scale and Diversity
> Thank you for raising this important point. While our dataset consists of 3,373 high-resolution, real-world images, making it the larger datasets for hybrid sensor ISP tasks now.
> For a more detailed response, please refer to "Summary and Answers of Official Reviews (2/2)."
>
> Additionally, we plan to maintain and expand this dataset as a dynamic, growing resource for the research community. We aim to enhance its diversity while ensuring its focus on high-quality, real-world data.
>
> ### Q.2 Reproducibility and Dataset Usage
> We greatly appreciate your suggestion regarding early release. However, to comply with double-blind review policies, we are unable to release the dataset during the review process. We are fully committed to making the dataset, benchmark code, and results publicly available immediately upon acceptance of the paper. This timeline ensures fairness and accessibility for the research community.
>
> ### Q.3 Event-Guided ISP Baseline Evaluation
> Thank you for this constructive suggestion. As the first dataset specifically designed for event-guided ISP tasks, our primary goal was to establish a baseline by demonstrating the utility of a simple event fusion approach.
> **Prior to this work, no research has explored using events to guide the RAW ISP process.**
>
> We recognize the importance of further developing and evaluating more event-guided architectures. To address this, we have included discussions in the revised manuscript to emphasize future directions, including advanced event-guided ISP methods and architectures.
>
> ### Q.4 Dataset Scale and Diversity:
> Thank you for this question. As mentioned, our dataset is already larger than other similar datasets and sufficient for training and testing ISP models effectively.
>
> We are also committed to expanding its scale and diversity over time by incorporating additional data, scenes, and lighting conditions. This dynamic approach will address current limitations and better serve the research community.
>
> ### Q.5 Early Data Release:
> Thank you for raising this question. To ensure accessibility and reproducibility, we will release the dataset, benchmark codes, and experimental results immediately upon acceptance of the paper. This timeline ensures compliance with double-blind review policies while supporting the broader research community.
>
> ### Q.6 Event-Guided ISP Baseline Evaluation:
> As this is the first dataset designed for event-guided ISP tasks, we demonstrated the potential of a simple baseline approach in this work.
>
>
> Once again, we sincerely thank you for your constructive feedback. Your comments have been invaluable in helping us improve the paper and refine its contributions. We hope our responses address your concerns and demonstrate the robustness of our work.
>
> Sincerely,

---

> ### Author Response · Authors · 2024-11-25
> **Looking forward to further discussions with reviewer qDHH.**
>
> Thank you for your time and attention. In the revision paper, we have elaborated on the dataset scale and provided additional examples to showcase its diversity. Additionally, we have committed to releasing the dataset and benchmark code upon acceptance of the paper. We hope you have had the opportunity to review our revisions and look forward to further engaging discussions with you.

---

> > ### Comment · Reviewer_qDHH · 2024-11-25
> >
> > Thank you for your explanations. I believe the primary contribution of this paper lies in the dataset. Therefore, I think it is essential to release the dataset and accompanying code to ensure clarity and accessibility for everyone. As such, I am inclined to maintain my original rating for now.

---

> > > ### Author Response · Authors · 2024-11-25
> > > **Ensuring Accessibility and Transparency: A Commitment to Open Data for Event-Based Vision Research**
> > >
> > > Thank you for your timely response.
> > >
> > > **We are delighted to see your positive recognition that the dataset is a major contribution of this paper.** Your acknowledgment is very encouraging to us.
> > >
> > > We assure you that **all datasets, along with the training and testing code, will be made publicly available to ensure clarity and accessibility for everyone**.
> > >
> > > Due to ICLR’s anonymous review policy, we are unable to share the dataset link with you directly at this stage. However, we will privately share the data with the Area Chair to ensure accessibility during the review process.
> > >
> > > Furthermore, we commit to making the dataset fully public immediately after the paper is accepted (upon lifting anonymity). **Aligned with ICLR’s OpenReview policies, we believe our commitment will also be closely observed by the event-based vision community.**
> > >
> > > We hope this clarifies your concerns and we look forward to further discussions.

---

### Official Review · Reviewer_q8r6 · 2024-11-13

**Soundness:** 3
**Presentation:** 2
**Contribution:** 2
**Rating:** 3
**Confidence:** 5

**Summary:**

This paper presents a event-RAW paired dataset of pixel-level aligned events and RAW images, including 3373 images across 24 scenes, for ISP process reforming. The study presents a conventional ISP pipeline that generates high-quality RGB frames for reference, performing basic ISP operations like demosaicing and white balancing. Some existing learnable ISP methods are trained and evaluated on the dataset.

**Strengths:**

1. The present paper introduces a event-RAW paired dataset that addresses a gap in the field of ISP and facilitates further research on event-guided ISP.
2. The trainable ISP methods are evaluated on this event-RAW paired dataset.

**Weaknesses:**

This paper, as a work primarily contributing a dataset, provides insufficient information about the dataset itself and focuses heavily on performance comparisons with existing methods, but the proposed method fails to outperform the current ones across all metrics. The overall logic and structure of the paper need improvement and refinement.
1. As a dataset-centric paper, it presents too few sample images, making it difficult for readers to intuitively grasp the specific content of the dataset and the differences in the scenes, camera shots, and exposure modes mentioned.
2. The introduction to the specific content of the dataset is insufficient. It is recommended to list statistical information about the dataset regarding the mentioned types and present them in tables.
3. The ISP presented in the paper is used to handle demosaicing, white balance, denoising, and color space transformations, but its capabilities in handling these tasks are not reflected in the performance analysis results.
4. Figures 6 and 7 have identical legends and can be combined into one figure, or one could be replaced with the visualization results of indoor data samples.
5. The experimental section only presents quantitative analysis results for outdoor and indoor data separately, failing to reflect the overall performance of the dataset.
6. The presented integrated improvement method does not enhance processing performance, and the paper does not provide sufficient explanation or analysis of this issue.
7. The experimental visual results are too few, and the layout is too compact, making it difficult for readers to intuitively appreciate the value of the dataset.
8. It is recommended to analyze the performance of different types of samples within the dataset using the same method in future work, rather than focusing primarily on comparing the performance of multiple ISP methods.

**Questions:**

1. The ISP method presented in this paper demonstrates varying effects across different tasks. It would be beneficial to analyze whether these effects are reflected in the proposed dataset.
2. The authors should explain and analyze the reasons why the proposed method fails to surpass existing methods in performance, as well as highlight its advantages.

---

> ### Author Response · Authors · 2024-11-19
>
> Dear Reviewer q8r6,
>
> Thank you for your insightful review and for highlighting the significant contributions of our work. We are especially grateful for your recognition of:
> (1) Dataset Contribution.
> (2)Thoroughly assessing trainable ISP methods on the dataset, showcasing its potential.
> Below, we address your concerns in detail, incorporating your valuable suggestions to improve the clarity and quality of our work.
>
> ### Q.1. Insufficient Dataset Information
> We appreciate your suggestion. To address this, we have:
> - (1) Added a detailed description of the dataset in Appendix Section B, including information on the types of scenes, exposure modes, and camera settings used.
> - (2) Included more visual samples to better illustrate the dataset's diversity and highlight specific scenarios of interest.
> We hope these updates will enhance readers' understanding of the dataset's content and significance.
>
>
> ### Q.2. Statistical Dataset Information
> Thanks for your suggestions. We have included a statistical summary table in the supplementary materials. This table provides a clear breakdown of the dataset by scene type, exposure mode, and camera-specific settings, offering a more comprehensive understanding of its structure and diversity.
>
>
> ### Q.3.Performance Analysis of ISP Tasks
> Thanks for your suggestions.
> We would like to clarify that the controllable ISP framework used in this work relies on ColorChecker-based evaluations to validate the accuracy of the generated RGB images, as shown in Figure 5.
> While intermediate processes such as demosaicing and white balancing are difficult to evaluate quantitatively due to the absence of ground truth, the final evaluation results provide an overall assessment of the pipeline's performance.
>
> For the learning-based methods, we have included comprehensive evaluations using objective metrics (e.g., PSNR, SSIM, L1) as well as non-reference metrics (e.g., NIQE, PI), which collectively demonstrate the effectiveness of our dataset in supporting ISP research.
>
>
> ### Q.4. Figures 6 and 7
> Thank you for pointing this out. In the revised manuscript:
> Figures 6 and 7 have been combined to reduce redundancy.
> Indoor data visualization has been added to Figure 8, providing a more diverse set of examples.
>
> ### Q.5. Comprehensive Dataset Performance Analysis
> We have updated the supplementary materials to include a combined evaluation of indoor and outdoor data. This comprehensive analysis provides a more holistic view of the dataset’s overall performance, addressing your concern.
>
> ### Q.6. Integrated Improvement Method
> Thank you for highlighting this. The event-guided ISP baseline introduced in our work is intended as a simple starting point to demonstrate the potential of integrating event data into ISP tasks. While this approach has shown performance gains in outdoor scenarios, it faces challenges in indoor conditions due to factors such as artificial lighting flicker. These findings, along with their implications, have been discussed in greater detail in the revised manuscript and supplementary materials.
>
> ### Q.7. Experimental Visual Results and Layout
> Thank you for highlighting this. We have significantly expanded the visual results in the supplementary materials. This includes:
> Adding more examples, particularly for low-light and fast-motion scenarios, as shown in Figure 10.
> Adjusting the layout in the main paper to ensure that visual results are presented more clearly and intuitively.
>
> ### Q.8. Sample-Specific Analysis
> We appreciate this suggestion. In response, we have conducted additional analyses in the supplementary materials to evaluate the performance of different dataset subsets. This provides deeper insights into the dataset's characteristics and potential applications.
>
> ### Q.9. Task-Specific Effects in the Dataset
> Thank you for raising this question. In the supplementary materials (Section E), we have added analyses discussing task-specific effects, such as the impact of lighting flicker in indoor scenes. These findings provide valuable insights into the dataset's strengths and limitations in supporting diverse ISP tasks.
>
> ### Q.10. Comparison with Existing Methods
> We acknowledge that the proposed method is a baseline approach designed to demonstrate the utility of the dataset rather than achieve state-of-the-art performance. The method involves a simple integration of event data into UNet and serves as a foundation for future research. We have included additional discussions in the supplementary materials analyzing its limitations, such as challenges in modeling global context, and its potential for further improvement.
>
> Once again, we thank you for your constructive feedback. Your comments have been instrumental in refining the quality and clarity of our work. We hope our responses address your concerns comprehensively and look forward to your further insights.
>
> Sincerely,

---

> ### Author Response · Authors · 2024-11-25
> **Looking forward to further discussions with reviewer q8r6.**
>
> Thank you for your valuable suggestions. In the revision paper, we have incorporated additional visual examples to intuitively showcase the specific content of the dataset. We have also improved the formatting to make the paper's structure more logical and accessible. Furthermore, we have expanded discussions on dataset scale and background knowledge. We hope you have had the chance to review these updates and look forward to engaging in further discussions with you.

---

> ### Author Response · Authors · 2024-12-01
> **Looking Forward to Further Discussions with Reviewer q8r6**
>
> Dear Reviewer q8r6,
>
> Thank you again for your valuable comments. We have carefully addressed your feedback and updates into the revision paper.
> **We sincerely hope to know if there are any unresolved concerns and look forward to engaging in further discussions with you.**
>
> Specifically, we have made the following revisions to address your points:
>
> ### 1. Sec. B [Q.1, Q.2]:
>   - Added additional visual examples (Fig. 11 and Fig. 13) showcasing various dataset scenarios.
>   - Included qualitative analyses (Fig. 12) summarizing key elements, lighting conditions, and their distributions.
> ### 2. Sec. A [Q.3]:
>   - Expanded discussion on the imaging principles of events and frames, emphasizing the advantages of event data for ISP tasks.
>   - Clarified the evaluation of the controllable ISP framework, which relies on ColorChecker-based assessments (Fig. 5). While intermediate processes like demosaicing and white balancing lack ground truth for quantitative analysis, we provided an overall pipeline performance evaluation.
> ### 3. Fig. 6 and Fig. 7 [Q.4]:
>   - Revised these figures as per your suggestion to reduce redundancy and improve clarity.
> ### 4. Tab. 8 and Sec. F [Q.5]:
>   - Conducted a comprehensive quantitative analysis of overall dataset performance.
> ### 5. Sec. 5.1, 5.3, D, and F [Q.6]:
>   - Discussed the benefits and challenges of event-data fusion, including its advantages and limitations under artificial lighting.
> ### 6. Fig. 6, Fig. 7, and Fig. 9 [Q.7]:
>   - Improved layout and added visual examples to illustrate the unique strengths of hybrid vision sensors.
> ### 7. Fig. 6, Fig. 7, Fig. 19 and Fig. 20 [Q.8]:
>   - Showcased the performance of the same method under varying scenes, providing insights into dataset diversity.
> ### 8. Sec. 5 and E [Q.9]:
>   - Analyzed task-specific performance across indoor and outdoor scenes, with a focus on lighting conditions and event-data gains.
> ### 9. Sec. 5.1, 5.2, and F [Q.10]:
>   - Provided a simple baseline fusion method, discussing its potential gains and associated challenges for future event-based ISP research.
>
> **We sincerely appreciate your time and effort in evaluating our work and look forward to engaging in further discussions to clarify any outstanding concerns.**
>
> Best regards,
> ICLR-4440 Authors

---

### Author Response · Authors · 2024-11-19
**Summary and Answers of Official Reviews (1/2)**

# Summary and Answers of Official Reviews (1/2)

Dear Reviewers,

We sincerely thank all reviewers for their thoughtful feedback and constructive comments. We deeply appreciate the recognition of our contributions and the valuable suggestions provided to improve our work. Below, we summarize the key strengths of our paper as highlighted by the reviewers in the Table A. We are grateful for the reviewers' recognition of the novelty, practical contributions, and thorough analysis presented in our work. These acknowledgments motivate us to further improve the quality and impact of our research.

Table A: Recognized Contributions.
| **Contribution**    | **Reviewer** | **Official Review**      |
| ---------------------------------------- | ------------ | ----------------------------------------------------------------------------------------------------------------------------------------------- |
| **1. Novel Research Question**    | vrds  | "Event-guided ISP seems like a **novel and interesting** idea."    |
|       | fVak  | "The paper presents a **new** event-RAW paired dataset..."  |
|       | AvAS  | "The relevant background knowledge of this paper is **clearly explained**."      |
|       |       | "The topic of RGB-event ISP is a very **interesting and meaningful** topic for future cameras."       |
|       | q8r6  | "This paper **addresses a gap** in the field of ISP and **facilitates further** research on event-guided ISP."      |
| **2. New Real-world Dataset**     | vrds  | "A **novel sensor** is designed to capture the datasets..."        |
|       | fVak  | "This paper introduces the RGB-Event ISP dataset and benchmark, **a novel paired dataset** that combines RAW and event data."     |
|       |       | "The dataset **fills an existing gap** by providing paired RAW and event data, the **first** of its kind specifically designed for ISP tasks."  |
|       |       | "The proposed ISP pipeline is **flexible**, making it versatile for testing various ISP algorithms."  |
|       | AvAS  | "I **greatly appreciate** the effort to create a dataset for RGB-Event ISP, which **opens up opportunities** for event-assisted RGB ISP tasks." |
|       | qDHH  | "This is the **first dataset** providing avenues for developing new ISP algorithms."    |
|       |       | "This paper presents a novel dataset **across various** scenes, exposure modes, and lenses."   |
| **3. Conventional ISP Pipeline**  | vrds  | "A **conventional ISP** pipeline is proposed to generate RGB references..."      |
|       | fVak  | "The authors design a **customizable ISP** pipeline, allowing for benchmarking..."      |
|       | AvAS  | "Using a controllable ISP pipeline developed by the authors, **high-quality RGB frames** are generated."     |
|       | qDHH  | "This work generated **high-quality RGB** images as ground truth by using a ColorChecker."     |
|       | q8r6  | "The study presents a conventional ISP pipeline that generates **high-quality RGB** frames for reference."   |
|       |       | "A **conventional ISP** pipeline is proposed and learnable ISP methods are used."       |
| **4. Thorough Analysis of Benchmarking** | vrds  | "The **analysis and results** look good."     |
|       |       | "The experiments **demonstrate the effectiveness** of several methods in both outdoor and indoor scenes."    |
|       | fVak  | "The experimental **evaluation is thorough**."       |
|       |       | "The paper also **addresses its limitations** and **suggests potential solutions**."    |
|       | qDHH  | "This work evaluates the proposed dataset by **testing various ISP** baseline methods along with an event-guided ISP approach."   |
|       |       | "Some **conclusions and insights** can be drawn from these experiments."  |
|       | q8r6  | "The trainable ISP methods are evaluated on this event-RAW paired dataset."      |

We sincerely appreciate the reviewers' valuable feedback and constructive suggestions, which have guided us in refining our work. In response to your insightful comments, we have carefully addressed each point and made substantial updates to the paper. These revisions have significantly strengthened the robustness and clarity of our research. Thank you again for your support and suggestions, which have greatly enhanced the quality of our paper.

Best

ICLR-4440 Authors

---

### Author Response · Authors · 2024-11-19
**Summary and Answers of Official Reviews (2/2)**

Thank you to all the reviewers for their hard work and careful review. Here we answer the questions of common concern.

## Concern about Dataset Size for ISP tasks:

We acknowledge the reviewer’s concern regarding the limited scale of our dataset. Below, we provide a detailed explanation to justify the sufficiency and significance of our dataset for ISP tasks:

### 1. ISP Tasks Focus on Pixel-Level Data·
Unlike traditional perception tasks, such as face detection, which often require datasets with tens of thousands of examples, ISP tasks focus on processing pixel-level data.
In ISP, every pixel with different neighbors can be considered an example, making the requirements of dataset size fundamentally different.
Therefore, our dataset is not only sufficient in terms of quantity (3,373) but also provides high-resolution (2248 × 3264) samples that are effective for training and testing ISP models.

### 2. Size Comparison with Related Dataset
Our work is inspired by the MIPI Demosaic Workshop [a], which represents the most closely related study in this field. Compared to the MIPI dataset, our dataset is significantly larger, containing over 3,373 real images. This is four times the size of the MIPI dataset, which includes only 800 images. Moreover, our dataset offers higher-resolution images, with dimensions of 2248 × 3264 pixels (approximately 7.3 million pixels per image), while the MIPI dataset has resolutions around 2K, such as 2040 × 1356 pixels.

Although the MIPI dataset is only about one-fourth the size of ours, it is still sufficient to train large networks, such as transformers [b]. Our dataset also supports more comprehensive training and testing.

More importantly, the MIPI dataset is entirely composed of simulated data [a], whereas our dataset is based on real-world data, providing a more realistic representation for ISP tasks. Additionally, our dataset includes authentic event streams, enabling research into event-guided ISP, which was not possible with the MIPI dataset.

Furthermore, the representative works [c,d,e] in ISP is summarized in Table B. These datasets all contain no more than 200 training samples. However, they feature high-resolution images, which are sufficient to support effective model training and evaluation.

In summary, in ISP tasks, the size of the dataset is not only related to the number of images but also to their resolution. Our dataset is sufficient in both aspects.

Table B, Size Comparison of Related Datasets
| Dataset | Resolution | Scalse | Real-World | Events | Tasks| Publication |
| --- | --- | --- | --- | --- | --- | --- |
| Ours | 2248 × 3264 | 3373 | Yes | Yes | hybrid sensor ISP | |
| MIPI[a,b] | 2K, e.g. 2040 × 1356 | 800 | No | No | hybrid sensor ISP | CVPR 2024 |
| ISPW[c] | 1368 × 1824, 4480×6720 | 197 | Yes | No | ISP | ECCV 2022 |
| NR2R[d] | 3464×5202 | 150 | Yes | No | ISP | CVPR 2022 |
| DeepISP [e] | 3024×4032 | 110 | Yes | No | ISP | IEEE TIP 2018 |



### 3. Diversity in Scenes and Conditions
Our dataset includes a wide variety of lenses, scenes, and lighting conditions. This diversity supports comprehensive testing across different scenarios, ensuring the dataset’s relevance and utility for a broad range of ISP tasks.

To further substantiate the sufficiency of our dataset, we have included a new section in the revised appendix of our paper.
This table highlights the scale, diversity, and resolution of our dataset compared to existing ISP-related benchmarks.

### 4. Long-term Maintenance
Our dataset provides a comprehensive resource for ISP research. Its real-world nature distinguishes it from existing datasets and makes it particularly well-suited for event-guided ISP tasks. Moreover, we are committed to the long-term maintenance of this dataset and plan to expand it in the future to accommodate larger and more complex tasks.


## Reference

- [a] Wu, Yaqi, et al. "MIPI 2024 Challenge on Demosaic for Hybridevs Camera: Methods and Results." Proceedings of the IEEE/CVF Conference on Computer Vision and Pattern Recognition. 2024.
- [b] Xu, Senyan, et al. "DemosaicFormer: Coarse-to-Fine Demosaicing Network for HybridEVS Camera." Proceedings of the IEEE/CVF Conference on Computer Vision and Pattern Recognition. 2024.
- [c] Shekhar Tripathi, Ardhendu, et al. "Transform your smartphone into a dslr camera: Learning the isp in the wild." European Conference on Computer Vision. Cham: Springer Nature Switzerland, 2022.
- [d] Li, Zhihao, Si Yi, and Zhan Ma. "Rendering nighttime image via cascaded color and brightness compensation." Proceedings of the IEEE/CVF Conference on Computer Vision and Pattern Recognition. 2022.
- [e] Schwartz, Eli, Raja Giryes, and Alex M. Bronstein. "Deepisp: Toward learning an end-to-end image processing pipeline." IEEE Transactions on Image Processing 28.2 (2018): 912-923.

---

### Author Response · Authors · 2024-11-19
**Summary of Revision Paper**

Dear Reviewers,

We thank all reviewers' thorough and careful evaluations. Here, we provide a summary of the revisions made to our paper.

**Please note that due to file size limitations, we have compressed the PDF document, which may result in blurry images. However, the original, clear PDF can be downloaded from the supplementary materials.**

Below, we outline the key revisions made to the paper.

**Section A**

We have added a new section titled "HYBRID SENSOR IMAGING PROCESS, PRINCIPLES, AND POTENTIAL."
- (1) Firstly, we included the imaging principles to help readers better understand the sensor's details.
- (2) Next, we analyzed the advantages of event data through theoretical analysis and practical demonstrations. For example, in Figure 9, we present cases of fast motion and low-light scenes.
- (3) Moreover, we examined the time alignment in imaging, which addresses the issue of synchronizing the two types of data.
- (4) Finally, we explored the potential of our approach in various ISP tasks, providing insights for future research.

**Section B**

We have added more dataset examples and a scale analysis.
- (1) To address the reviewers' concerns, we included additional examples from our dataset.
- (2) Furthermore, we analyzed the size of our dataset and compared it with studies most similar to ours. Compared to the latest research, our dataset is four times larger.
- (3) We also compared our dataset with classic ISP methods, demonstrating that its scale is sufficient to support training.
- (4) **Diversity Testing with GPT-4o**: We conducted a comprehensive analysis of dataset diversity using ChatGPT-4. This includes statistical evaluations of captured scenes, objects, lighting conditions, and weather variations, offering a clearer understanding of the dataset's richness.

**Section E**

We added a discussion on how hyperparameters affect model training results.

**Section F**

We included a discussion of the results of various methods on the entire dataset.

Best

ICLR-4440 Authors

---

### Meta-Review · Area_Chair_MiXd · 2024-12-20

**Metareview:**

The paper introduces an event-RAW paired dataset and an event-guided ISP pipeline, addressing a clear gap in the ISP domain. The reviewers acknowledged the significance of the dataset and the approach but raised concerns about the dataset's scale, reproducibility, and the method's limited performance gains. The authors' revisions addressed these concerns effectively by expanding dataset details, clarifying event integration, and providing additional evaluations. While some limitations remain, the work lays a good foundation for future research in event-guided ISP. The AC recommends acceptance for its valuable contribution to the community and potential to inspire further advancements.

**Additional Comments On Reviewer Discussion:**

Reviewers raised concerns about dataset scale, reproducibility, clarity of event integration, and performance of the proposed method. The authors addressed these by expanding dataset descriptions, improving clarity, adding visual examples, and explaining alignment processes. Reviewer q8r6 gave the most critical rating. During the rebuttal period, the AC determined that the authors adequately addressed Reviewer q8r6's concerns. Despite multiple reminders, Reviewer q8r6 provided no further feedback. A few reviewers raised concerns about the dataset's size and/or availability. The AC concluded that the dataset is still significantly larger than existing benchmarks and is confident that the authors will release it as promised. The rebuttal effectively clarified key concerns, leading to a final decision favoring acceptance for the paper's foundational contribution.

---

### Decision · Program_Chairs · 2025-01-22

Accept (Poster)